Application of nanotechnology in fruit crops—from synthesis to sustainable packaging

Ramya S. 1
Auxcilia J. 1 auxcilia@tnau.ac.in
http://orcid.org/0000-0002-5285-6578 Paital Biswaranjan 2
Sharmila D. Jeya Sundara 3
Vethamoni P. Irene 1
Venugopal Sheela 4
http://orcid.org/0000-0002-6584-0030 Indra N. 1
Subramanian Kizhaeral S. 3
http://orcid.org/0000-0003-3748-3395 Sahoo Dipak Kumar 5 dsahoo@iastate.edu
1 Department of Fruit Science, Tamil Nadu Agricultural University , Coimbatore, Tamil Nadu , India
2 Redox Regulation Laboratory, Department of Zoology, College of Basic Science and Humanities, Odisha University of Agriculture and Technology , Bhubaneswar, Odisha , India
3 Centre for Agricultural Nanotechnology, Tamil Nadu Agricultural University , Coimbatore, Tamil Nadu , India
4 Centre for Rice, Tamil Nadu Agricultural University , Coimbatore, Tamil Nadu , India
5 Department of Veterinary Clinical Sciences, College of Veterinary Medicine, Iowa State University , Ames, Iowa , United States
Singh Anshuman
Electronic publication date: 2025 Jun 23
Publication date: 2025
Volume: 13
Electronic Location ID: e19603
Received 2024 Dec 4; Accepted 2025 May 23
Copyright: © 2025 Ramya et al.
Copyright year: 2025
Copyright holder: Ramya et al.
License: This is an open access article distributed under the terms of the Creative Commons Attribution License, which permits unrestricted use, distribution, reproduction and adaptation in any medium and for any purpose provided that it is properly attributed. For attribution, the original author(s), title, publication source (PeerJ) and either DOI or URL of the article must be cited.
License URL: https://creativecommons.org/licenses/by/4.0/

Keywords: Nanocoatings, Nanofertilizers, Nanopackaging, Nanopesticides, Nanosynthesis, Precision farming, Resilience, Environmental sustainability

Funding: Department of Biotechnology, Govt of India, New Delhi This research received no specific grant from any funding agency in the public, commercial, or not-for-profit sectors. The Star College Scheme, funded by the Department of Biotechnology, Govt of India, New Delhi, was awarded to Biswaranjan Paital. The funders had no role in study design, data collection and analysis, decision to publish, or preparation of the manuscript.

==============================
Fresh fruits, rich in essential nutrients and bioactive compounds, contribute positively to human health. However, their perishable nature and post-harvest shelf life contribute to significant worldwide losses, posing sustainable challenges in quality preservation and reducing waste in fruit production. Thus, many advancements have been developed, including nanotechnology, which can potentially increase fruit production by improving its quality, efficiency, and sustainability. Nanoscience is rapidly advancing as one of the key areas of applied research, offering diverse applications in fruit crops. Nanoparticles used in the form of nano-fertilizers, nano-pesticides, nano-coatings, nanofilms, and nano packaging have distinct features used for targeted site-specific pest and disease management, smart nutrient supply, and delivery via biosensor(s) in fruit crops. Moreover, they are synthesized efficiently, functioning rapidly in a cost-effective and environmentally sustainable manner. These innovations collectively address critical challenges in fruit crop management, including promoting plant growth and stress resilience, boosting productivity, extending shelf life, reducing post-harvest damage, and improving crop quality while mitigating environmental impact and ensuring food safety. This review comprehensively highlights substantial insights into using nanoparticles as a promising technique for increasing fruit crop resilience and ensuring food security in the context of environmental changes, as well as the recent application of nanotechnology at various stages of fruit production.

Introduction

Fruit crops play a crucial role in the global economy, contributing to agricultural trade, employment, and rural development. As consumer demand for fresh and processed fruits continues to rise, countries with favorable climates and production capabilities benefit from high export revenues (Gergerich et al., 2015). The fruit industry supports farmers and supply chain workers and drives logistics, food processing, and biotechnology advancements. Beyond economic significance, fruits are essential to human nutrition due to their rich composition of vitamins, minerals, fiber, and antioxidants (Abobatta, 2021). The growing awareness of health benefits has increased the preference for organic and minimally processed fruits, further shaping global agricultural practices and trade policies. Despite their importance, the international fruit industry faces numerous challenges, such as climate change, weather patterns, pest infestations, post-harvest losses, and market fluctuations, threatening fruit production and profitability (Bhattacharjee et al., 2022). Additionally, the overuse of chemical pesticides and fertilizers has raised environmental concerns (Sah et al., 2024), leading to stricter regulations and consumer demand for sustainable farming practices (Beyuo et al., 2024). To address these challenges, conventional management strategies include integrated pest management, efficient post-harvest handling, and storage technologies, which help to reduce losses and maintain fruit quality. In recent years, frontier technologies such as nanotechnology have led to innovative solutions for mitigating these hazards (Manzoor et al., 2024). Nanotechnology is an emerging strategy for increasing fruit productivity with limited inputs in contemporary fruit cultivation (Kamatyanatti et al., 2019). Nanoscience is the study of materials at the nanoscale (109 m) from 1–100 nanometers (Singh, 2017). Nanomaterials have unique physical and chemical properties that differ from those of conventional materials larger than 100 nanometers (Kumar et al., 2024). Nanoparticles have unique chemical and physical qualities that promote plant growth, development, and stress tolerance (Fig. 1), making them helpful in improving fruit crops (Manzoor et al., 2024). Nanomaterial seed coatings have attracted significant interest in fruit crops due to their ability to enhance plant growth, increase crop yields, and improve resource efficiency. Nanomaterial coatings help seeds adhere better to the soil, reduce wastage during planting, and boost planting efficiency (Silvestre, Duraccio & Cimmino, 2011). Recently, nanoparticles have improved plant tolerance against biotic and abiotic stresses. Nanoparticles play a crucial role in enhancing plant yield characteristics under stress conditions. It significantly affects various physiological processes, including stress response mechanisms, hormone metabolism, osmolyte biosynthesis, ethylene production, and signaling pathways (Rasheed et al., 2022).

Figure 1 Role of nanotechnology in fruit crops.

A tree graph representing the significance of nanotechnology in fruit cultivation. It has been reviewed that nanotechnology has multidimensional use in the agriculture fields, starting from farming to post-harvest management of crops. As a result an increased productivity shall be obtained in cropping plants. Various nano-based products are utilized in fruit crops. Disease management and safety storage of post-harvested crops are the most challenging issues in agriculture. So, the use of nano-products such as nanofertilizers, nano pesticides, and nanofungisides is used even in post-harvest packaging.

Nanomaterials provide numerous beneficial functions in biological systems; nevertheless, their toxicity can also be demonstrated to be detrimental (Paital, 2020; Jena et al., 2022; Yadav et al., 2023). Therefore, green synthesis of nano-particles and nano-herbals is now being used to open a new horizon in all fields, including horticulture, either to protect the crops or to use their products as neutraceuticals, crop protectors, herbicides, pesticides, etc., (Wesley et al., 2014; Paital, 2020; Ilango et al., 2022; Patel et al., 2023, 2025; Mishra et al., 2024; Subaramaniyam et al., 2025). So, organizing information and their critical evaluation of the role of nanomaterials on organisms is essential. Pests, such as insects, mites, nematodes, and diseases, significantly impact crop profitability (Kroumova et al., 2013; Sahoo et al., 2014, 2017, 2021; Reddy, 2015; Yoon et al., 2018). Using pesticides frequently has led to insect and disease resistance, accumulating residues in produce, and environmental damage (van Bruggen, Gamliel & Finckh, 2016; Patel et al., 2024). As a result, alternative pest and pathogen control strategies are required. Nanotechnology has the potential to effectively manage insects and pathogens through targeted pesticide delivery and early detection systems (Rana et al., 2024). The most frequent nanomaterials in fruit production include packaging, nano-insecticides, nano-fertilizers, nano-fungicides, and precision fruit culture (Rana et al., 2021). Nanoparticles are highly stable and biodegradable, making them suitable for producing nanocapsules to carry insecticides, fertilizers, and other agrochemicals. Nanoparticles’ slower release of functional molecules limits their use in many applications (Hassan, Al-Hchami & Alrawi, 2020). Nanoparticles perform differently than bulk particles due to their smaller size, higher charge, larger surface area, and increased stability and solubility (Shrestha, Wang & Dutta, 2020). Recently, focus has been given heavily to producing bio-based edible coverings to improve the post-harvest processing longevity of fruits. Added to that, nanotechnology has been recognized as an excellent approach (Travičić, Cvanić & Ćetković, 2023) for increasing coating qualities, a better moisture barrier, and superior mechanical, optical, and microstructural capabilities, as well as the progressive and controlled discharge of bioactive substances. Some nanotechnology-based plant extracts are frequently used to extend the post-harvest shelf life of fruits.

Fruits coated with edible nanocoating have an extended shelf life as they effectively retain moisture and preserve their freshness. This is due to the coating’s protective layering, which keeps gases and water vapour from entering or exiting the fruit and preserves its texture, colour, and firmness (Sharma et al., 2024). These coatings improve barrier qualities on the outer covering of fruits, creating a favorable microenvironment by optimizing the concentration and impeding the ripening process. A diverse spectrum of nano-based precision and tiny equipment, which includes nano-sensors (Mishra et al., 2017), nano-based gadgets, machines, and robotics, is used in modern fruit production. These nanomaterial-based biosensors are also used in high-tech fruit production. Nano-biosensors play a vital role in transforming farming by developing diagnostic tools. These sensors are accurate, reliable, and economical in dealing with various agricultural, food, and environmental concerns (Dar, Qazi & Pirzadah, 2020). Some agricultural sensor uses include identifying heavy metal ions, contaminants, microbial load, and pathogens, and monitoring temperature, traceability, and humidity. Nanotechnology has transformed fruit packaging by improving the functionality of traditional materials to safeguard perishable goods. Nanoparticles, including silver (AgNPs), zinc oxide (ZnO), titanium dioxide (TiO₂), and nanoclays, are commonly employed to create active packaging films with excellent antibacterial, antioxidant, and gas barrier capabilities. These nano-enhanced coatings can suppress the growth of spoilage bacteria and minimize oxidation, significantly increasing fruit shelf life (Silvestre, Duraccio & Cimmino, 2011). For example, silver nanoparticles integrated into packaging materials have shown high antibacterial action against common fruit diseases, hence delaying deterioration during storage. The growing demand for sustainable alternatives to typical plastic packaging has sparked great interest in biodegradable nanomaterials such as nanocellulose, chitosan, and starch-based composites, which improve food preservation while minimizing environmental impact. These materials not only have longer shelf lives due to their antibacterial qualities and controlled release of preservatives, but they also break down more effectively than conventional plastics (Lefèvre et al., 2006; Motelica et al., 2020). With the current context of improved crop growth and yield using nano-fertilizers, nano-pesticides, nano-biosensors for soil health, the target pest and disease management using nanoparticle-based biocides and nano-carriers for bio-pesticides, post-harvest preservation and shelf-life extension of fruits using nano-coatings, antimicrobial packaging, ethylene control methods, for quality enhancement of the processed fruit and their products using nano-emulsions for flavor and nutrient enhancement, improved texture and stability, for the detection of contaminants and quality monitoring using nanosensors, etc., nano-science can lead to the reduced chemical usage and with less environmental impacts in one hand and increase in precision and efficiency with improved product quality and safety on the other hand. So, the use of nano-technology in the challenges and considerations, including safety and toxicity in fruits and fruiting crops, reduced cost and scalability, regulatory approval, etc., needs to be reviewed on a priority basis. Therefore, it is suggested that nanotechnology holds transformative potential for managing fruiting crops, pre- and post-harvest quality handling of fruits, and their derived products, specifically for extending shelf life. This review article thoroughly highlights significant insights into the application of nanoparticles as a promising method for enhancing fruit crop resilience and ensuring food security amid environmental changes, along with the recent use of nanotechnology at different stages of fruit production.

Methods of literature review

A thorough search was carried out across major databases such as PubMed, Science Direct, Web of Science, Scopus, Agricola, and Google Scholar, with relevant terminologies (Oza et al., 2024; Doshi et al., 2024) such as “fruit crops and nanotechnology” were added with additional terms such as challenges, harvest, post-harvest, shelf life, texture, packaging, quality, scalability, safety, environmental impacts, regulatory, transport, fertilizer, pesticide and soil health. The inclusion criteria concentrated on peer-reviewed studies published in the recent decade, with a specific emphasis on the use of nanotechnology in fruit production and post-harvest management. Key data, including aims, techniques, and outcomes, were gathered and organized into categories. Articles merely containing the search words but out of the scope of the topic were rejected. Articles in English that fall under the topic were screened, and >200 articles were selected for the review in an unbiased method. Articles were selected irrespective of specific laboratory, person, or country of publication. Each study was critically appraised for quality and relevance, identifying gaps, limitations, and areas for further research.

Synthesis of nanomaterials

Nanomaterials, nanoparticles, and nanoemulsions play a significant role in transforming agricultural practices, especially in fruit crops (Avestan, Naseri & Najafzadeh, 2018; Hmmam et al., 2021; Basumatary et al., 2021; Khan et al., 2023; Singh et al., 2024; Thakur et al., 2024; Daler et al., 2024). The synthesis of nanoparticles involves techniques like sol-gel processes, chemical vapor deposition, and biological methods using plant extracts or microorganisms for eco-friendly production (Atanda, Shaibu & Agunbiade, 2025). Nanomaterials, produced through mechanical milling or self-assembly methods, are also integrated into the packaging to extend fruit shelf life and reduce post-harvest losses (Leta, Adeyemi & Fawole, 2024). Furthermore, nanosensors, synthesized via thin-film deposition techniques, aid in monitoring plant health and soil conditions, enabling precision agriculture (de Oliveira Filho et al., 2021).

Nanoemulsions, synthesized through high-energy techniques like ultrasonication or low-energy methods like phase inversion temperature, offer innovative solutions for fruit crops (Sneha & Kumar, 2022). These nanoemulsions act as edible coatings enriched with antioxidants and antimicrobial agents to maintain fruit quality, delay spoilage, and enhance marketability (Thakur et al., 2024). Their controlled release properties improve the delivery of essential bioactive compounds, such as nutrients and protective agents, ensuring improved fruit texture, appearance, and nutritional value (Akonjuen & Aryee, 2023). By addressing challenges like microbial contamination and water loss, these nanotechnology-based solutions significantly contribute to sustainable agriculture and the global fruit supply chain (Ahmad et al., 2024) (Fig. 2, Table 1).

Figure 2 Methods of synthesis of nanoparticles used in fruit crops.

Nanoparticles employed in fruit crops are manufactured utilizing physical, chemical, and biological processes, with benefits in terms of scalability, stability and environmental compatibility. Their size defines their mode of application, which might be foliar spraying, soil integration, or seed coating. These nanoparticles work through various processes, which includes regulated release of active chemicals, increased nutrient absorption, and targeted disease and pest management.

Table 1 Method of synthesis, mode of delivery, and role of nanoparticles in fruit crop.

Method of synthesis	Size range	Mode of application	Fruit crop	Mode of action	References	
Co-precipitation method (copper nanoparticles)	10–50 nm	Foliar spray, soil amendment	Banana (Musa sp.)	Resistance against fusarium wilt, improved yield	Kumar et al. (2024)	
Electrochemical method (silver nanoparticles)	10–50 nm	Edible coating	Mango (Mangifera indica)	Reduced microbial spoilage, extended shelf life	Hmmam et al. (2021)	
Co-precipitation method (iron nanoparticles)	20–100 nm	Invitro	Apple (Malus domestica)	Improved growth and nutrient uptake	Avestan, Naseri & Najafzadeh (2018)	
Wet chemical method (zinc oxide nanoparticles)	20–80 nm	Foliar spray	Strawberry (Fragaria ananaasa)	Inhibited fungal growth, improved quality	Singh et al. (2024)	
Solvo thermal method (titanium dioxide nanoparticles)	5–20 nm	Edible coating	Peach (Prunus persica)	Improved UV protection and shelf life	Khan et al. (2023)	
Ionic gelation method (chitosan nanoparticles)	50–200 nm	Edible coatings, foliar spray	Pineapple (Annanas comosus)	Reduced microbial activity, prolonged freshness and extended shelf life	Basumatary et al. (2021)	
Sol-gel method (silicon nanoparticles)	5–100 nm	Soil amendment	Grapes (Vitis vinifera)	Enhanced nutrient uptake, stress tolerance	Daler et al. (2024)	
Nanoemulsions	50–200 nm	Edible coating	Citrus fruits (Citrus sp.)	Prolonged freshness, microbial reduction	Thakur et al. (2024)	

Nanomaterial—seed coating

In fruit crops, the application of nanomaterials in seed priming is an emerging research area aimed at improving seed performance by supplying nutrients, biostimulants, enhancing seed germination and seedling growth (Shukla et al., 2019). Nanomaterials influence germination, yield, and stress tolerance by modulating gene expression, optimizing plant metabolism, and improving nutrient uptake, promoting better plant development (Zaman, Ayaz & Park, 2025). Nanoscale seed coatings in fruit crops, using materials like ZnO and SiO2, form a protective barrier that enhances germination, improves nutrient and water uptake, ensuring early seedling development (Shelar et al., 2023). One of the primary advantages of using nanomaterial seed coatings in fruit crops is their capacity to protect seeds from environmental stressors such as pests, diseases, and harsh weather (Zhao et al., 2024). Acting as a barrier, these materials safeguard seeds during their most vulnerable stages, leading to higher germination rates and the development of healthier, more resilient plants. Furthermore nanoparticles can contain vital nutrients, growth regulators or beneficial microorganisms, allowing for targeted and regulated release to seedlings, which improves root development, stress tolreance and overall plant vigor throughout early growth phases in fruit crops such as papaya, pomegranate, citrus and other seed propagated fruit crops (Abdelmigid et al., 2022). This targeted delivery ensures that plants obtain the necessary resources for vigorous growth and robust development. By enhancing nutrient absorption and promoting beneficial microbial interactions, these coatings contribute to improving crop vitality and yield (Mahra et al., 2025). In addition, nanomaterial coatings help seeds adhere better to the soil, reducing wastage during planting and boosting planting efficiency—a critical factor in horticulture where optimal seed spacing and placement are essential for successful crop development. While the potential benefits of nanomaterial seed coatings are substantial, it is crucial to use them responsibly, considering both safety and regulatory guidelines (Zaim et al., 2023). When applied appropriately and within regulatory frameworks, nanomaterial seed coatings could transform the practices by improving crop quality, increasing yields, and promoting sustainable, efficient cultivation methods in fruit crops.

Nanofertilizers—salutary role in fruit crops

Nanofertilizers, an emerging innovation in agriculture, offer a proper solution to improve nutrient efficiency, productivity, and sustainability in fruit crops (Kumar et al., 2023; Zagzog & Gad, 2017; Roshdy & Refaai, 2016; Davarpanah et al., 2016; El-Hameed et al., 2017; Abdel-Hak et al., 2018; Abdelaziz et al., 2019; Ranjbar, Ramezanian & Rahemi, 2020; Akbar et al., 2019; Shalan, 2020; Elsheery et al., 2020; Zahedi et al., 2021; Aly et al., 2022). Nano-fertilizers have several advantages over conventional fertilizers, as these substances are harmless and less harmful to the natural world and humans (Sharma et al., 2021). Nano-fertilizers can be derived from various plant parts using physical, chemical, mechanical, or biological techniques, or they can be synthesized from modified forms of traditional fertilizers (Gade et al., 2023) to improve soil fertility, productivity, crop quality standards, and lower expenses while raising profits (Fig. 3). Nano-fertilizers can prepare one or more plant nutrients to boost growth and production while performing better (Harith Burhan Al Deen Abdulrhman et al., 2021), using less fertilizer and releasing nutrients more slowly than conventional fertilizers (Table 2).

Figure 3 Role of nanofertilizers and shelf-life in fruit crops.

Several pieces of evidence fortifying the idea of the use of nano-fertilizers are clear. Less amount of use with cheap price and high efficiency are the main advantages. Positive impacts of nanofertilizers on tree growth and development, as well as soil health, have been documented. It increases the resistance capacity of plants along with better growth. Factors affecting the shelf life of fruits after harvest can also be influenced by nanomaterials. Usually, ripened fruits are more prone to damage during transport, sorting, and grading. Microbial activity and environmental factors can also enhance the degrading process. Nanomaterials can be used at each stage to protect the post-harvested fruits.

Table 2 Beneficial role of nanofertilizers in various fruit crops.

Fruits	Variety	Nanofertilizers	Properties	References	
Apple (Malus domestica)	Red delicious	Nano calcium	Quantitative and qualitative character	Ranjbar, Ramezanian & Rahemi (2020)	
Grapes (Vitis vinifera)	Flame seedless	Nano fertilizers (amino-minerals, orgland active-Fe, Boron-10, Amino-Zn, Super-Fe)	Improved berry colouration and high fruit quality	Wassel, El-Wasfy & Mohamed (2017)	
Grapes (Vitis vinifera)	Flame seedless	carbon nano-tubes (CNTs) from total nitrogen	Increased leaf area, leaf fresh weight and leaf dry weight, shoot length, shoot diameter and number of leaves per shoot of grapevines	Abdel-Hak et al. (2018)	
Apple (Malus domestica)	Anna	Ag and Zn nanofertilizer	Increased total chlorophyll content, fruit set percentage, fruit yield, fruit’s physical and chemical characteristics	Aly et al. (2022)	
Mango (Mangifera indica)	Kiette	Nanoboron	Increased shoot length, thickness, leaf area, and number of leaves per shoot.	Abdelaziz et al. (2019)	
Grapes (Vitis vinifera)	Crimson seedless	Nano-powder potassium sulfate	Leaf area, internode length	Shalan (2020)	
Pomegranate (Punica granatum)	Malase saveh	Nano-Se	Higher leaf NPK content	Zahedi et al. (2019)	
Strawberry (Fragaria ananassa)	Queen elisa	Nano-silicon oxide	Salt tolerance	Akbar et al. (2019)	
Strawberry (Fragaria ananassa)	Chandler	Nano zinc	Increased number of leaves	Kumar et al. (2017)	
Mango (Mangifera indica)	Ewais	Nano-ZnO and Si	Salt stress tolerance	Elsheery et al. (2020)	
Mango (Mangifera indica)	Zebda & Ewasy	Nano zinc	Highest number and weight of fruits, total tree yield, and percentage of TSS in fruits, Reduced malformation	Zagzog & Gad (2017)	
Pomegranate (Punica granatum)	Ardestani	Nano-iron and Nano-Boron	Number of fruits, iron content of leaves, total sugars, and the total yield	Davarpanah et al. (2016)	
Datepalm (Phoenix dactlylifera)	Zaghloul	Nano NPK	Higher fruit yield, bunch weight, total soluble solids, total sugars and pulp percentage	Roshdy & Refaai (2016)	

Nanoparticles enhance the efficiency of nutrient uptake and the overall quality of fruits (Zahedi, Karimi & Teixeira da Silva, 2020). Additionally, it has been put forth that balanced fertilization of agricultural produce can be accomplished by nanotechnology. Nanoparticles boost plant development by resisting infectious diseases and plant solidity by preventing bending and causing deeper rooting of crops (Dharam Singh et al., 2017). This technology has enabled the exploitation of small nanomaterial particles carried on the fertilizer to build the so-called smart fertilizer, which enhances the efficiency of nutrient use and reduces the costs of protecting the environment by intelligently controlling the speed of nutrient release (Tarafdar et al., 2015) to match the absorption pattern of crops and improving the solubility of insoluble nutrients in the soil, it reduces its adsorption and stability and increases its availability.

Nanoparticles—their role in mitigating abiotic stress of fruit crops

Abiotic stress has globally imposed environmental issues, which have a significant impact that leads to a reduction in the production and productivity of fruits (Dilnawaz, Misra & Apostolova, 2023). Nanotechnology plays a substantial role in mitigating abiotic stress in fruit crops, as nanoparticles have shown positive effects on plants under abiotic stress conditions (Zarafshar et al., 2015; Nava et al., 2017; Cosme Silva et al., 2017; Zahedi et al., 2019, 2021; Orooji et al., 2020; Wang et al., 2021; Mahmoud et al., 2021; Mahmoudi et al., 2022; Hassan et al., 2022; Tejada-Alvarado et al., 2023), as they can be used to assist plants in coping with abiotic stress management (Khalid et al., 2022). Nanoparticles infiltrate plants through their roots and leaves, causing biochemical, morphological, molecular, and physiological changes in crops during stress. Nanoparticles have significant effects on various physiological processes, including stress response mechanisms, hormone metabolism, osmolyte biosynthesis, ethylene production, and signaling pathways involving nitric oxide, abscisic acid (ABA), and calcium. They also regulate signal transduction pathways during drought and salinity stress, activating stress-responsive genes to enhance plant survival (Rasheed et al., 2022). Nanoparticles play a crucial role in improving plant yield under drought and salinity conditions. They help mitigate water loss by maintaining water balance, ultimately improving abiotic stress tolerance. Nanoparticles also regulate stomatal conductance and transpiration rates by influencing leaf anatomy and promoting stomatal closure (Acosta-Motos et al., 2017). Additionally, nanoparticles protect photosynthetic machinery, enhance photosynthesis, and activate antioxidant systems to repair damage caused by reactive oxygen species (ROS) in chloroplasts and photosystems. Furthermore, they stimulate the electron transport chain and increase chlorophyll content in plant cells (Forni, Duca & Glick, 2016; Manzoor et al., 2022) (Table 3). Overall, the application of nanoparticles is essential for helping plants withstand drought and salinity, maintaining their normal functions, promoting environmental health, and sustaining crop yield.

Table 3 Role of nanoparticles in mitigating abiotic stress in fruit crops.

Fruits	Nanoparticles	Properties	References	
Strawberry (Fragaria ananaasa)	Se-NPs	Tolerance to salinity, and subsequently yield, which were attributed to their ability to protect photosynthetic pigments	Zahedi et al. (2019)	
Pomegranate (Punica granatum)	Se-NPs	Fruit cracking caused by drought stress was reduced	Zahedi et al. (2021)	
Banana (Musa sp.)	Chitosan-NPs	Improve plant resilence to chilling injury—suitable in cold affected regions, Serves as osmoprotectant	Wang et al. (2021)	
Mango (Mangifera indica)	Chitosan-NPs	Retards the senescence process	Cosme Silva et al. (2017)	
Sweet orange (Citrus sinensis)	Sio2-NP	Tolerant to salt stress	Mahmoudi et al. (2022)	
Strawberry (Fragaria ananaasa)	Fe3O4 NPs	Decreased level of H2O2	Orooji et al. (2020)	
Grapefruit (Citrus × paradisi)	ZnO-NPs	Photocatalytic activity	Nava et al. (2017)	
Pineapple (Annanas comosus)	Ag-NPs	Increase the content of pigments	Tejada-Alvarado et al. (2023)	
Pear (Pyrus pyrifolia)	SiO2-NPs	Si and K content increased	Zarafshar et al. (2015)	
Loquat (Eriobotrya japonica)	SiO2-NPs	Chilling tolerance	Wang et al. (2020)	
Olive (Olea europaea)	Nano-Si	Tolerant to water stress	Hassan et al. (2022)	
Plum (Prunus domestica)	Chitosan-Arginine NPs	Chilling tolerance	Mahmoudi et al. (2022)	

Nanopesticides—propitious effect on fruit crops

Nanotechnology is used extensively in plant protection to enhance crop yield (Moulick et al., 2020). Conventional crop protection methods often involve using large quantities of fungicides, herbicides, and insecticides. Approximately 90% of pesticides are ultimately lost in the environment or do not effectively reach their intended targets for pest control (Tudi et al., 2021). Having active chemicals at the right concentration in a formulation is of the utmost importance for protecting plants from pests and preventing crop loss. Agricultural research has focused on developing innovative plant protection formulations called nanoformulation, or pesticide encapsulation, that have transformed plant protection technology (Bhagat, Samanta & Bhattacharya, 2013; Rao & Paria, 2013; Hua et al., 2015; Young et al., 2018; Zhao et al., 2018; Sharma et al., 2021; Wu et al., 2023). Nanoformulation, often known as pesticide encapsulation, has transformed the plant protection sector. Nanoencapsulation of pesticides involves coating active ingredients with nano-sized materials; the materials (Yadav et al., 2021) that are encapsulated are called the coated nanomaterials’ internal phase, and the materials that are encapsulated are called the core material’s external phase (pesticides).

Pesticide encapsulations provide a controlled release of active ingredients into root areas or inside plants, all without impacting efficacy (Maluin & Hussein, 2020). Conventional pesticide or herbicide formulations, on the other hand, limit pesticide water solubility while also injuring other organisms, resulting in increased resistance to target organisms. For a sustainable agro-environmental system, nanomaterials in pesticide formulations provide advantageous properties such as improved durability, flexibility, stability under heat, solubility, crystallinity, and biodegradability (Chaud et al., 2021). Using active substances in a timely and controlled manner reduces the need for pesticides for pest and disease control (Table 4), an essential aspect of IPM. Sustainable agriculture requires minimal use of agrochemicals to prevent environmental degradation and harm to non-target species; thus, nano-pesticides sparingly minimize agricultural production costs (Shang et al., 2019).

Table 4 Effects of employing nanopesticides in fruit crops.

Fruits	Varieties	Nanopesticide	Pathogen	Mode of action	References	
Sweet orange (Citrus sinensis)	Pineapple	Nano-ZnO	Citrus canker	Fruit canker incidence reduced from 63% to 7%	Graham et al. (2016)	
Grapefruit (Citrus paradisi)	Ruby	Nano-CuO	Citrus canker	Fruit infection reduced to 25% from 60%	Young et al. (2018)	
Citrus (Citrus sp.)	Tankan	Nano-Calcium carbonate (CaCo3)	Oriental fruit fly	Insecticide—Damage caused by Oriental fruit flies decreased	Hua et al. (2015)	
Guava (Psidium guajava)		Insect pheromone nanogel	Fruit fly	Improved insects catch in the fly for insecticide formulation apparatus for nanogel formulation	Bhagat, Samanta & Bhattacharya (2013)	
Apple (Malus domestica)		Nano-sulphur	Apple scab	Fungicide—Inhibited 93% of the fungal growth	Rao & Paria (2013)	
Strawberry (Fragaria × ananaasa)		Nano-chitosan	Anthracnose	Fungicide	Wu et al. (2023)	

Nanocoatings

Increased consumer awareness regarding fresh fruits’ health and nutritional advantages has led to a consistent rise in their demand. However, due to their high moisture content, fruits are highly perishable, creating an ideal environment for the growth of pathogenic and spoilage microbes. This diminishes their shelf life and compromises safety and quality (Mohammad & Ahmad, 2024). Nanocoatings, thin films (<100 nm) applied to a substrate to enhance its properties and performance, offer notable benefits over traditional coatings. These include resistance to stains, antibacterial and antioxidant properties, odor management, and even distribution of active agents. In the fruit industry, nano-coating is frequently utilized in packaging applications. By integrating active bioactive ingredients, nanocoatings provide active food packaging with antibacterial and antioxidant features (Gago et al., 2020). Specific types of food packaging are coated with nanoparticles to enhance shelf life, security, and package quality (Fig. 4). Active packaging coatings, a promising technology in food packaging, utilize preservatives and nanocoatings to serve as antimicrobial, antifungal, and antibacterial agents, as well as protective coatings and self-cleaning surfaces for food contact (Souza et al., 2015; Li et al., 2011, 2021; Kittitheeranun, Dubas & Dubas, 2012; Arnon et al., 2014; Nadim et al., 2015; Salvia-Trujillo et al., 2015; Deng et al., 2017; Robledo et al., 2018; Prakash, Baskaran & Vadivel, 2020; Melo et al., 2020; Miranda et al., 2021, 2022; Kalia et al., 2021; Jafarzadeh et al., 2021; Ngo et al., 2021; Odetayo et al., 2022; Shi, Xiang & Jiahu, 2024) (Table 5). Using edible films containing nanocoatings to coat fruit products has made significant strides in recent years, enhancing food safety.

Figure 4 Role of nanocoatings and nano-packaging in fruit crops.

Post-harvested fruits become damaged under several conditions, and packaging and coating of fruits with compatible materials are a challenge from a health point of view. Therefore, nano-coatings are used to increase the self-life of ripened fruits. It also protects fruits against microbial damage. Nano-based packaging in fruit crops also is proposed to be used. Nano-based packaging enhances the self-life of post-harvested fruits, especially at their ripening stage. Rapid involvement and more research in this field are warranted.

Table 5 Nanocoatings and their properties in fruit crops.

Fruits	Nanomatrix and Bioactive compound	Property	References	
Apple–Fuji (Malus domestica)	Sodium alginate + Lemongrass oil	Antimicrobial activity	Salvia-Trujillo et al. (2015)	
Strawberry (Fragaria × ananaasa)	Chitosan + Thymol	Antimicrobial activity	Robledo et al. (2018)	
Papaya–Redtainung (Carica papaya)	Hydroxylpropyl methylcellulose + carnauba wax	Reduce moisture loss	Miranda et al. (2019)	
Pineapple (Ananas comosus)	Sodium alginate + citral	Increase in antimicrobial activity	Prakash, Baskaran & Vadivel (2020)	
Mandarin–Nova (Citrus reticulata)	Carnauba wax + oleic acid	Antimicrobial activity	Miranda et al. (2021)	
Pear–Barlett (Pyrus pyrifolia)	Chitosan + cellulose nanocrystal and oleic acid	Increased adhesion, delayed ripening	Deng et al. (2017)	
Mangoes (Mangifera indica)	Sodium alginate + chitosan	Firmness, microbial protection	Souza et al. (2015)	
Citrus (Citrus sp.)	Carboxymethy cellulose + chitosan	Enhanced fruit glossiness and prevented weight loss	Arnon et al. (2014)	
Mango (Mangifera indica)	Polystyrene sulfonate sodium salt + Poly diallyldimethyammonium chloride	Improved hydrophilicity of the surface	Kittitheeranun, Dubas & Dubas (2012)	
Strawberry (Fragaria × ananaasa)	Nanocomposite Zinc Oxide-Chitosan coatings + Polyethylene films	Increase quality and shelf life of fruit and antimicrobial activity	Jafarzadeh et al. (2021)	
Banana–Cavendish (Musa sp.)	Aloe vera and Moringa plant extract edible coatings + chitosan nanoparticles	Improved efficiency and increased the storage life of banana	Odetayo et al. (2022)	
Strawberry (Fragaria × ananaasa)	Methylcellulose-based edible coating	Maintenance of fruit quality during storage	Nadim et al. (2015)	
Strawberry (Fragaria × ananaasa)	Chitosan tripolyphosphate nanoparticles suspension	Acts as an antibacterial agent	Melo et al. (2020)	
Blueberry (Vaccinium corymbosum)	Chitosan	Delays mould and yeast formation	Li et al. (2021)	
Mango (Mangifera indica)	Nano-chitosan	Firmness of fruits	Ngo et al. (2021)	
Apple (Malus domestica)	nano-Zno	Increased shelf life by 6 days	Li et al. (2011)	
Peach (Prunus persica)	Bacillus circulans + Nano-ZnO	Enhanced shelf life	Shi, Xiang & Jiahu (2024)	
Guava (Psidium guajava)	Urticadiocia leaf extracts + Nano-ZnO, CuO	Enhanced shelf life of guava	Kalia et al. (2021)	

Nanocomposite materials

Nanocomposite materials encompass one-dimensional, two-dimensional, and three-dimensional components mixed at the nanometer scale. In contrast to conventional packaging materials, nanocomposites offer added advantages such as increased strength, enhanced biodegradability, and superior management of gaseous molecules (Rovera, Ghaani & Farris, 2020), crucial for the development of high-performing packaging materials (Kalia & Parshad, 2015). Typically, a nanocomposite material (Table 6) consists of three distinct components: the matrix material, filler, and filler interface material (Sharma et al., 2022), with at least one component at the nanoscale (Yang et al., 2010; Emamifar et al., 2010; Esmailzadeh et al., 2016; Fortunati, Mazzaglia & Balestra, 2019; Vieira et al., 2020a, 2020b; He et al., 2021; Kalia et al., 2021; La et al., 2021; Sun et al., 2021; Ezati, Riahi & Rhim, 2022).

Table 6 Nanocomposite-based packaging in fruit crops.

Fruits	Matrix + Nanoparticles	Microbistatic effect	Reference	
Strawberry (Fragaria × ananaasa)	LDPE + Silver and titanium dioxide nanoparticles	Aspergillus flavus	Yang et al. (2010)	
Orange juice (Citrus sp.)	Polyethylene + Silver and titanium dioxide nanoparticles	Aspergillus flavus	Emamifar et al. (2010)	
Pineapple Juice (Ananus comosus)	Polyethylene + Silver nanoparticles	Bacillus subtilis	Fortunati, Mazzaglia & Balestra (2019)	
Kiwi (Actinidia deliciosa)	Polyethylene + Silver nanoparticles	Bacillus subtilis	Fortunati, Mazzaglia & Balestra (2019)	
Grapes (Vitis vinifera)	Polyethylene + Silver nanoparticles	Bacillus subtilis	Fortunati, Mazzaglia & Balestra (2019)	
Apples (Malus domestica)	Nanoparticles	Enterobacterae rogenes	Esmailzadeh et al. (2016)	
Strawberry (Fragaria × ananaasa)	Cellulose nanocrystals + Silver	Escherichia coli	He et al. (2021)	
Cherries (Prunus avium)	Sodium alginate + Silver	Salmonella aureus & Escherichia coli	Sun et al. (2021)	
Papaya (Carica papaya)	HPMC + Silver	C. gloeosporioides	Vieira et al. (2020b)	
Banana (Musa sp.)	Chitosan + ZnO	Bacillus subtilis	La et al. (2021)	
Guava (Psidium guajava)	Chitosan + ZnO	Salmonella aureus	Kalia et al. (2021)	
Banana (Musa sp.)	Carboxymethyl cellulose + TiO2	Listeria monocytogenes	Ezati, Riahi & Rhim (2022)	

Nanopackaging

Nanotechnology has shown great promise in the food processing industry to improve post-harvest technologies that help prevent neglect and lower losses (Liu, Zhang & Bhandari, 2020). To address the worldwide issue of fresh product security, the farming sector should prioritize protecting fruits and vegetables (Ijaz et al., 2020). Controlling pre-harvest and post-harvest conditions can improve the shelf life of fresh fruit (Palumbo et al., 2022). The primary reason for adopting nano in food packaging is to improve the protective barrier qualities of packaging materials (Ghosh et al., 2025). Nano-based alimentary packaging materials also provide antibacterial properties, operate as oxygen scavengers, and act as moisture barriers (Rai et al., 2019).

Bio-based packaging

Bio-based packaging uses biodegradable films to regulate moisture transfer and gas exchange during the packaging of food goods. This improves safety and preserves nutritional and sensory quality. Such packaging supplies are considered more environmentally friendly than other standard packaging films (Chandra et al., 2020). Bio-based packaging protects food products from environmental factors such as microbes, relative humidity, and gas conditions. Biodegradable packaging films possess the ability to be broken down by living organisms, distinguishing them from other packaging options. This package type is seen as more environmentally friendly. Bio-based packaging encompasses improved, active, and smart packaging (Fig. 5) (Kuswandi, 2017).

Figure 5 Types of biobased nanopackaging system and the working model of nano-based fruit crop management.

Several modes of packaging are adapted to protect fruits from post-harvest damage. The use of nano-materials is suggested to improve post-harvest management. Working of nanosensors in fruit crops. Sensors transmit information about the tree’s condition, which is analyzed and passed along to the decision support system.

Active packaging

Nanomaterials are utilized in active packaging to improve product protection by directly interacting with the food or environment. Nano-silver, nano-copper oxide, nano-magnesium oxide, nano-titanium dioxide, and carbon nanotubes are expected to have potential use in antimicrobial food packaging (Agriopoulou et al., 2020). It is an oxygen-scavenging packaging with enzymes between polyethylene layers. Active packaging can prevent microbial development after opening and rewrapping using an active film (for example, antimicrobial film, oxygen scavenging films, and UV-absorbing films).

Improved packaging

Nanocomposites, which contain up to 5% w/w nanoparticles and clay nanoparticles (Arash et al., 2023), improve barrier properties (80–90% reduction) in packaging materials (e.g., nanocoating, nanolaminates, clay nanoparticles).

Smart packaging

Nanomaterials in smart packaging detect biochemical or microbiological changes in food, such as pathogens and spoilage gases (Onyeaka et al., 2022). Reactive particles in packing materials can provide information about the product’s status (such as nanosensors). Nanosensors act upon external stimuli to communicate, inform, and identify products, ensuring their quality and safety.

Precision farming in fruit crops

Nanomaterial engineering is a leading research field for sustainable agricultural development. Nanomaterials in precision agriculture minimize expenses, boost efficiency, and promote sustainable growth (Shang et al., 2019). Precision fruit culture is becoming increasingly crucial for assessing and tracking the growth of trees, soil parameters (moisture, nutrients, pH, EC, and so on), disease detection, pesticide penetration, and environmental impact using nanosensors. Precision fruit culture enhances fruit quality while ensuring the health of soil and plants, promoting ecological sustainability and environmental security (Longchamps et al., 2022). Nanomaterial engineering is used in high-tech fruit cultivation to provide a more specific surface area for the sustainable development system. The primary use of nano-fruit cultivation is to produce high-quality fruit with cheap input costs while maintaining ecological sustainability. In this culture, nanosensors, nanotechnology-based GPS, supercomputers, and remote sensing devices are used (Mittal et al., 2020).

Nanosensors

Nanosensors enable plants to communicate, making it more straightforward to understand dynamic changes in plants’ environment and physiological states. Nanosensors have been created to suit the demands of agricultural development. These sensors provide accurate and real-time monitoring of individual plants on a micro-scale with excellent temporal resolution (Giraldo et al., 2019). They also help to translate optical, wireless, and electrical signals into plant signaling molecules (Vurro et al., 2024). Nano-sensors and nano-biosensors have potential uses in the food industry, including monitoring food processing, quality assessment, packaging, storage, shelf life, food safety, microbial contamination, toxins, and residual contamination. Nanosensors are often designed for specific applications in food and agriculture (Srivastava, Dev & Karmakar, 2018). Nano-biosensors have the potential to be an extremely useful instrument for intelligent delivery systems, enhancing soil health, irrigation safety, pesticide detection, and plant pathology. Nano-biosensors can also detect seed viability, fruit shelf life, and plant nutrient requirements (Fig. 5). Furthermore, they play a crucial part in protecting crops and advancing the idea of sustainable agriculture. Nanoparticles, including gold, silver, and magnetic nanoparticles, graphene oxide, carbon nanotubes, and wireless nanosensors, have been used to improve sensing (Oerke et al., 2005; Fernández-Baldo et al., 2010; Shojaei et al., 2016; Tereshchenko et al., 2017; Dhiman et al., 2019) (Table 7). Commercializing nanosensors requires substantial intellectual property and patent rights to ensure long-term viability.

Table 7 Types of nanosensors used in fruit crops.

Fruits	Nanosensors	Detection	References	
Grapes (Vitis vinifera)	ZnO-based films	Grapevine virus A-type (GVA) proteins (GVA-antigens)	Tereshchenko et al. (2017)	
Citrus (Citrus sp.)	cdTe quantum dots Nanocarbon dots	Fluorometric immunoassay-Citrus tristeza virus	Shojaei et al. (2016)	
Apple–Malus domestica
Pears–Pyrus pyrifolia
Grapefruit–Citrus × paradisii	Carbon based screen printed electrode	Plum pox virus	Fernández-Baldo et al. (2010)	
Apple–Malus domestica	IR thermography (DIRT)	Apple scab	Oerke et al. (2005)	
Citrus (Citrus sp.)	Microfluidic electrochemical immunosensor (nanochip)	Yellow shoot disease (Huanglongbing)	Dhiman et al. (2019)	

Conclusion

Presently, a lot of technological innovation is being developed and utilised at various phases of fruit production. One such innovation is nanotechnology which has the potential to increase fruit yield with diminished farm risks and has a more comprehensive application such as nano-fertilizers, nano-pesticides, nano-coatings, post-harvest dips, packaging, increasing water use efficiency, and plant defense measures, all of which play essential roles in boosting the development of plants, improving reproductive growth, and blossoming, thus increasing efficiency, the quality of the product, shelf-life, and reducing fruit waste. Nanomaterials are utilized for targeted site-specific pest and disease management, targeted and slow nutrient supply (smart delivery), and pest and disease detection in fruit crops via biosensor delivery (Fig. 6). Nanomaterials are quick, inexpensive, and environmentally friendly. They may be developed quickly, with minimal effort, and without affecting the environment. The application of nanoparticles in fruit production has the potential to revolutionize it, enhancing productivity while minimizing resource input. The application of nanoparticles in fruit production holds considerable promise for enhancing sustainable and precise fruit production in developing countries.

Figure 6 A schematic presentation of application of nanotechnology in management of fruiting crops and their associated products.

Future perspectives and challenges

Nanotechnology offers tremendous potential to transform fruit cultivation by enhancing productivity, quality, and sustainability. The recent innovations in nanotechnology include nano-fertilizers, nano-pesticides, nano-coatings, nanosensors, nanopackaging, and other nanomaterials like carbon nanotubes, silica nanoparticles, and biodegradable nano-coatings derived from polymers such as chitosan. Nanotechnology also facilitates the early detection of pests and diseases using nanosensors and enhances plant resistance through advanced delivery systems. Post-harvest management includes nano-coatings that prolong the shelf life of the fruits, smart packaging, and technologies that regulate the ripening process. Additionally, nanotechnology promotes sustainable agriculture by reducing inputs, improving water use efficiency, and stress management in fruit crops. Integrating nanosensors with smart farming enables real-time monitoring of soil, water, and nutrients. However, challenges such as high production costs, regulatory barriers, and environmental safety need to be addressed to ensure safe and effective implementation of nanotechnology in fruit crops. By overcoming these limitations, nanotechnology provides innovative solutions to enhance fruit crop productivity and sustainability by addressing the growing demands of global food systems.

Supplemental Information

Supplemental Information 1 Various applications of nanotechnology in horticulture, especially in fruit crops.

Examples include nano-seed coating for improved germination and stress resistance, nanopackaging for increased shelf life and food safety, nanoparticles for targeted delivery systems, nanosensors for real-time crop health monitoring, nanopesticides for controlled pest management, and nanofertilizers for efficient nutrient delivery. Collectively, these advances represent a sustained strategy aimed at enhancing agricultural productivity, resource efficiency, and ensuring environmental protection.

We are grateful to the Professor and Head, Department of Fruit Science, Dean, Horticultural College and Research Institute, TNAU, Dean (SPGS), TNAU, Coimbatore.

Additional Information and Declarations

Competing Interests

The authors declare that they have no competing interests.

Author Contributions

S. Ramya performed the experiments, analyzed the data, prepared figures and/or tables, authored or reviewed drafts of the article, and approved the final draft.

J. Auxcilia conceived and designed the experiments, performed the experiments, analyzed the data, prepared figures and/or tables, authored or reviewed drafts of the article, and approved the final draft.

Biswaranjan Paital conceived and designed the experiments, performed the experiments, analyzed the data, authored or reviewed drafts of the article, and approved the final draft.

D. Jeya Sundara Sharmila performed the experiments, analyzed the data, prepared figures and/or tables, authored or reviewed drafts of the article, and approved the final draft.

P. Irene Vethamoni conceived and designed the experiments, analyzed the data, authored or reviewed drafts of the article, and approved the final draft.

Sheela Venugopal conceived and designed the experiments, analyzed the data, authored or reviewed drafts of the article, and approved the final draft.

N. Indra conceived and designed the experiments, analyzed the data, authored or reviewed drafts of the article, and approved the final draft.

Kizhaeral S. Subramanian conceived and designed the experiments, analyzed the data, authored or reviewed drafts of the article, and approved the final draft.

Dipak Kumar Sahoo conceived and designed the experiments, performed the experiments, analyzed the data, authored or reviewed drafts of the article, and approved the final draft.

Data Availability

The following information was supplied regarding data availability:

This is a literature review.

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
