# Peer review of "Application of nanotechnology in fruit crops—from synthesis to sustainable packaging"

_PeerJ, doi:10.7717/peerj.19603_

## Round 0.1 · original submission · Major Revisions

Dear Dr. Sahoo,

Thank you for your submission to PeerJ.

It is my opinion as the Academic Editor for your article - Use of nanotechnology in management of fruiting crops and their associated products - that it requires a number of major and minor revisions before being accepted for publication.

You are therefore advised to carefully go through each and every comment raised by the reviewers and modify your manuscript in accordance with these suggestions. In addition to addressing the reviewers comments, Please also take into account the queries while revising your manuscript.

Editor's comments

1. The manuscript title may be changed to “Use of nanotechnology in management of fruit crops and their associated products

2. The authors are advised to better articulate their view point in relation to various facets of nano-technology application in fruit crops. They should first broadly describe the importance of fruit crops in global economy, their growing importance in human nutrition, challenges facing global fruit industry, some of the conventional management options to address those challenges, and finally to elaborating the role of frontier technologies including nano-technology in mitigating these hazards. Therefore, they will have to thoroughly revise the Introduction section.

3. Under the head ‘Methods of literature review’ (lines 126-135), authors have described the method of literature review. However, there are no supporting references.

4. While discussing the role of nano-technology in fruit crops in the subsequent sections of the manuscript, the authors should try to maintain coherence and a proper sequence, i.e., from propagation to post-harvest value chain that involves different facets such as planting, initial care, pest and diseases management, etc. up to harvesting and storage/transportation. Authors should try to incorporate relevant information on the use of nano-tech in such areas.

5. The role of nano-technology and nano-formulations in mitigating the abiotic stresses in fruit crops is entirely missing. Accordingly, authors are exhorted to present and 3-4 paragraphs on these applications in perennial fruit crops. Insights may also be taken from other perennial woody crops, if not directly available from fruit crops.

6. Did authors use any specific tool(s) for drawing the illustrations presented in various figures. Some pictures seem to have been obtained from internet, and may not be advisable to be used in light of copyright violations and potential plagiarism. Therefore, it may be looked into.

7. Some Tables (2, 4 and 5) are too short to draw any reasonable conclusions. Therefore, authors should present some additional findings in these tables to improve their quality.

You may accordingly revise your manuscript and submit it for additional peer reviews.

Good luck!

Reviewer 1 ·

Basic reporting

.

Experimental design

.

Validity of the findings

.

Additional comments

This paper entitled Use of nanotechnology in management of fruiting crops and their associated products found some question in this paper so it needed for major revision:

• In the introduction section, the authors should incorporate recent findings and additional information related to nanotechnology.
• The authors must adapt the figures in the paper to align with those from previously published studies. Additionally, they should include information on the application of nanoparticles (NPs) for managing abiotic stress for fruit crop and other crop too in table and other place.
• To improve the clarity and comprehension of the research, the authors should include a graphical representation illustrating the review article.

·

Basic reporting

The review is indeed interesting and addresses a highly relevant topic. However, I have the following comments and suggestions to improve the clarity, depth, and overall quality of the manuscript:

1. Introduction:
The introduction should be expanded to provide a clearer research gap and specific objectives. Currently, the introduction offers a broad overview, but it would benefit from a more focused discussion on the gaps in current knowledge and how your review aims to address them.
Please clearly state the objectives of the review to guide the reader through the scope of the paper and highlight the unique contribution this review is making to the field.
2. New Subtitle:
Add a new title in the manuscript: "Advancements in Nanotechnology for the Management of Fruiting Crops and Enhancement of Their Associated Products".
To strengthen this section, please cite the following literature and any other relevant recent studies:
https://doi.org/10.3390/suschem5020004
https://doi.org/10.1007/s42452-024-06265-7

This will help connect your review with broader sustainability goals and emphasize the environmental and economic importance of nanotechnology.
3.All figures should have clarity and self explanatory. Please ensure that all figures are discussed in the appropriate sections of the text. This will help readers understand the relevance and context of each figure.
4. Importance of Nanotechnology in fruit crops and their products:
Please include the following literature to enhance this section:
https://doi.org/10.1021/acsomega.2c01400
https://doi.org/10.1002/slct.201904826
https://doi.org/10.1007/s42452-024-06265-7

These studies provide additional insights and should be integrated into the discussion to give a more robust explanation of the significance of advancements in nanotechnology for the management of fruiting crops and their associated products.
5. Permissions for Figures:
Please ensure that you have obtained the necessary permissions for any figures taken from previous literature. Securing permission for figures is mandatory to avoid any copyright issues and to comply with publication guidelines.
6. Language and Grammar:
There are several areas in the manuscript where English language corrections are needed. A thorough proofreading or grammar check will be beneficial to improve the clarity, flow, and professionalism of the manuscript.
7. Future Perspectives:
After the conclusion, please add a section on "Advancements in Nanotechnology for the Management of Fruiting Crops and Enhancement of Their Associated Products". This section should focus on the future advancements in technology, potential applications, and the challenges that remain in the field.

Experimental design

Appropriate. However, some relevant and recent references can also be added for enrichment of article.

Validity of the findings

Article have findings but need to be included more research findings.

Additional comments

The introduction should be expanded to provide a clearer research gap and specific objectives. Currently, the introduction provides a broad overview, but it would benefit from a more focused discussion on the gaps in current knowledge and how your review aims to address them.

Clearly state the objectives of the review to guide the reader through the scope of the paper and highlight what unique contribution this review is making to the field.

·

Basic reporting

The manuscript is well written.
There are a few grammatical errors which the author can improve.
Clarity is there.
Background of the subject is written good.
Introduction is also clear.

Experimental design

Article content is within the Aims and Scope of the journal.
At some places It is observed that the paragraph should contain more information specially under those headings where paragraph is too short.
Methods used in the review are ok.
In the first line of introduction, it is written nanoscale, and in bracket it is written 10 to the power 9 which is wrong...the correct one is 10 to the power -9.
At some places Tab is there to start a paragraph whereas at some places Tab is not there.. so uniformity should be maintained.

Validity of the findings

The manuscript is not very much novel but yes it contains the meaningful information related to the topic of the manuscript.
Conclusion is well stated.
Figures are good.
Table also contain valuable information.
Reference formatting is not uniform. Maintain unformity.

Additional comments

Although the manuscript is fine, but still some more novel information can be added to the manuscript to enhance the quality of the manuscript.
I find the manuscript little bit short.. This could be more elaborate. hence, some more information can be added as it is a review article.
Graphical abstract can be added.

---

## Round 0.2 · Major Revisions

Dear Dr. Sahoo

Thank you for your submission to PeerJ.

It is my opinion as the Academic Editor for your article - Use of nanotechnology in management of fruit crops and their associated products - that it requires some revision.

The reviewers have highlighted a range of major and minor shortcomings in your manuscript. Therefore, it can not be accepted in the present form. You are advised to carefully go through each and every comment and revise the manuscript critically to improve its quality. It is pertinent to mention that many comments and queries raised in the previous version were not addressed. This time you are expected to place equal emphasis on each and every query and suggestions including those raised by the Editor as well. In case, they remain unanswered, it may not be possible for me to consider your submission any further. I would also like to inform you that your revised submission will undergo additional peer review to ensure that the changes made by you are in sync with the recommendations.

Good luck

Reviewer 1 ·

Basic reporting

The author has revised the manuscript well but needs to include additional information:
1. The author should provide details on the type and synthesis of nanoparticles (NPs), accompanied by a flow diagram.
2. The author should explain how NPs/nanofibers (NFs) are applied, including the mode of application, and either create or adapt a relevant figure from previous studies.
3. Include a new table summarizing the type of NPs, their size, mode of application, and corresponding results.

Experimental design

NA

Validity of the findings

NA

·

Basic reporting

Dear author,
Please see the guidelines for manuscript writing and references style as it is not corrected yet after revision. So, i am recommending for major revision.

Experimental design

No comments

Validity of the findings

It is good finding. However, there is still corrections needed. After major revision, it can only be accepted for publication.

·

Basic reporting

Reframe the title of the paper.
Please clarify the main objective of the study and hypotheses early in the introduction.
The references cited in the MS are not in format kindly correct them.
Clarify which tests were used and justify why they were appropriate for the data.
The manuscript (MS) lacks a proper baseline comparison; add discussion to validate the findings.
Correct the entire spacing problem throughout the MS.
Write the keywords in alphabetical order.

Experimental design

In materials and method section, include specific parameters, controls, or experimental conditions that allow replication of the study.
The tables and figures are not clearly referenced in the main text. Kindly discuss appropriately.
The captions of the figure are too brief or incomplete. Also recheck all the table number.
The order of the result is confusing. So, restructure the result part with the observation.
The discussion part lacks the depth of the study. Kindly provide the insight into how the results compare with previous studies and explain any differences.
Add a section discussing how the results can be applied or influence the field.
The manuscript lacks a clear logical flow. There are multiple grammatical errors. Kindly re-read and correct them.
The limitation of the study is not acknowledged.

Validity of the findings

There is repetition in some sections, especially in the results and discussion.
The conclusion part is weak. Kindly summarize the key findings, discuss their importance, and suggest future directions clearly.
The references are not according to the journal’s format. Kindly go through the journal and format accordingly.
Also, italicize the scientific names in the reference part.

---

## Round 0.3 · Minor Revisions

Dear Dr. Sahoo,

Thank you for your submission to PeerJ.

It is my opinion as the Academic Editor for your article - Application of nanotechnology in fruit crops – from synthesis to sustainable packaging - that it requires a few Minor Revisions.

This decision is based on the fact that one of the reviewers is not in agreement with some parts of the manuscript, and wants them to be revised in order to make it accepted. Therefore, you are advised to follow their instructions and modify the manuscript in due course.

Hope to receive the revised manuscript shortly.

·

Basic reporting

Dear Authors,

Thank you for your efforts in revising the manuscript. Most of the corrections have already been incorporated, but there are still a few areas that need further improvement:

Abstract: Please provide a concise yet comprehensive summary of the entire content of the research paper. The abstract should clearly highlight the primary objectives, methodology, key results, and conclusions. A well-written abstract should include the background or context, the main problem or question being addressed, the methods employed, the key findings, and the broader implications or significance of the work. Kindly revise the abstract to reflect these elements.

Introduction: This review highlights the packaging of fruits using nanotechnology and its impact on fruit longevity. Please enhance the introduction by adding relevant content on this topic, incorporating references to support the claims.

Figures and Diagrams: The quality of the figures and diagrams is not up to the required standard. Please revise and improve them for better clarity and resolution.

Additionally, as this review addresses an evolving field, it would be beneficial to include further insights on emerging trends in nanotechnology. Specifically, the use of biodegradable nanomaterials and new regulatory frameworks for the safety and approval of nanomaterials in food packaging would add significant value to the manuscript. Kindly include these aspects with relevant references.

Experimental design

No major revisions needed regarding study design or evidence backing. A few slight enhancements to the forward-looking perspective would make it even sharper.

Validity of the findings

No comment

Additional comments

Picture quality should be improved and remade again for clarity.

·

Basic reporting

All earlier comments has been resolved properly hence MS can be accepted in the present form

Experimental design

Fine
All earlier comments has been resolved properly hence MS can be accepted in the present form

Validity of the findings

Fine

Additional comments

No

---

## Round 0.4 · accepted · Accept

Dear Dr. Sahoo,

Thank you for your submission to PeerJ.

I am writing to inform you that your manuscript - Application of nanotechnology in fruit crops – from synthesis to sustainable packaging - has been Accepted for publication.

Congratulations!

·

Basic reporting

All the comments has been resolved properly.

Experimental design

All the comments has been resolved properly.

Validity of the findings

All the comments has been resolved properly.

Additional comments

NA